# Low-Dimensional Compounds Containing Bioactive Ligands. Part XIX: Crystal Structures and Biological Properties of Copper Complexes with Halogen and Nitro Derivatives of 8-Hydroxyquinoline

Martina Kepeňová [1], Martin Kello [2], Romana Smolková [3], Michal Goga [4], Richard Frenák [4], Ľudmila Tkáčiková [5], Miroslava Litecká [6], Jan Šubrt [6] and Ivan Potočňák [1,*]

1 Institute of Chemistry, P. J. Šafárik University in Košice, Moyzesova 11, 041 54 Košice, Slovakia
2 Department of Pharmacology, P. J. Šafárik University in Košice, Trieda SNP 1, 040 11 Košice, Slovakia
3 Department of Ecology, Faculty of Humanities and Natural Sciences, University of Prešov, Ulica 17. Novembra 1, 081 16 Prešov, Slovakia
4 Department of Botany, Faculty of Science, Institute of Biology and Ecology, P. J. Šafárik University in Košice, Mánesova 23, 040 01 Košice, Slovakia
5 Department of Microbiology and Immunology, University of Veterinary Medicine and Pharmacy, Komenského 73, 041 81 Košice, Slovakia
6 Centre of Instrumental Techniques, Institute of Inorganic Chemistry of the CAS, Husinec-Řež č.p. 1001, CZ-25068 Řež, Czech Republic
* Correspondence: ivan.potocnak@upjs.sk; Tel.: +421-55-234-2335

**Abstract:** Six new copper(II) complexes were prepared: $[Cu(ClBrQ)_2]$ (**1a**, **1b**), $[Cu(ClBrQ)_2]\cdot 1/2$ diox (**2**) (diox = 1,4-dioxane), $[Cu(BrQ)_2]$ (**3**), $[Cu(dNQ)_2]$ (**4**), $[Cu(dNQ)_2(DMF)_2]$ (**5**) and $[Cu(ClNQ)_2]$ (**6**), where HClBrQ is 5-chloro-7-bromo-8-hydroxyquinoline, HBrQ is 7-bromo-8-hydroxyquinoline, HClNQ is 5-chloro-7-nitro-8-hydroxyquinoline and HdNQ is 5,7-dinitro-8-hydroxyquinoline. Prepared compounds were characterised by infrared spectroscopy, elemental analysis and by X-ray structural analysis. Structural analysis revealed that all complexes are molecular. Square planar coordination of copper atoms in $[Cu(XQ)_2]$ (XQ = ClBrQ (**1a**, **1b**), BrQ (**3**) and ClNQ (**6**)) and tetragonal bipyramidal coordination in $[Cu(dNQ)_2(DMF)_2]$ (**5**) complexes were observed. In these four complexes, bidentate chelate coordination of XQ ligands via oxygen and nitrogen atoms was found. Hydrogen bonds stabilizing the structure were observed in $[Cu(dNQ)_2(DMF)_2]$ (**5**) and $[Cu(ClNQ)_2]$ (**6**), no other nonbonding interactions were noticed in all five structures. The stability of the complexes in DMSO and DMSO/water was evaluated by UV-Vis spectroscopy. Cytotoxic activity of the complexes and ligands was tested against MCF-7, MDA-MB-231, HCT116, CaCo2, HeLa, A549 and Jurkat cancer cell lines. The selectivity of the complexes was verified on a noncancerous Cos-7 cell line. Antiproliferative activity of the prepared complexes was very low in comparison with cisplatin, except complex **3**; however, its activity was not selective and was similar to the activity of its ligand HBrQ. Antibacterial potential was observed only with ligand HClNQ. Radical scavenging experiments revealed relatively high antioxidant activity of complex **3** against ABTS radical.

**Keywords:** copper complexes; derivatives of 8-hydroxyquinoline; crystal structure; bromination; biological properties

## 1. Introduction

Cancer treatment by using cisplatin spread widely after discovery of its anti-neoplastic activity [1]. Since then, some derivatives, such as carboplatin, oxaliplatin, nedaplatin, heptaplatin, lobaplatin and miriplatin, were prepared and used as anticancer agents, too [2]. Nowadays, nearly half of all cancer diseases are being treated by platinum-based drugs. These compounds are exceptionally successful against a broad spectrum of cancers, which

is caused by their high reactivity [3]. Unfortunately, due this property, some negative phenomena are observed. Treatment with platinum-based drugs negatively influences healthy cells, which causes side-effects, such as neurotoxicity, renal toxicity, vomiting, and damage to the gastrointestinal tract, hair follicles and other tissues [4]. Another observed negative impact is the development of resistance, by which cancer cells try to escape apoptosis [5]. The necessity to reduce the negative impacts widely opened a new research field focused on other metal complexes as biological agents.

The quinoline family, including 8-hydroxyquinoline (8-HQ) and its derivatives, represents compounds with interesting pharmacological properties. For these compounds, anticancer, antibacterial, antifungal, antimalaria, antineurodegenerative and antiHIV effects were described [6–8]. These ligands can be coordinated to different metal atoms by oxygen and nitrogen atoms, and the resulting complexes often show increased anticancer activity. As an example, we can mention several complexes of Pd [9,10], Zn [11,12], Ga [13–15], Ru [16,17] and lanthanides [18–21], among which complexes with halogen- and nitro-derivatives of 8-HQ exhibited the highest activity. However, information on the anticancer activity of copper complexes rarely appears in the literature [12,22–25]. Therefore, we decided to prepare a series of copper complexes with commercially unavailable halogen- and nitro-derivatives of 8-HQ (HClBrQ = 5-chloro-7-bromo-8-hydroxyquinoline, HClNQ = 5-chloro-7-nitro-8-hydroxyquinoline and HdNQ = 5,7-dinitro-8-hydroxyquinoline), as well as with the commercially available, but hitherto unstudied HBrQ (HBrQ = 7-bromo-8-hydroxyquinoline) ligand: [Cu(ClBrQ)$_2$] (**1a**, **1b**), [Cu(ClBrQ)$_2$]·1/2 diox (**2**), [Cu(BrQ)$_2$] (**3**), [Cu(dNQ)$_2$] (**4**), [Cu(dNQ)$_2$(DMF)$_2$] (**5**) and [Cu(ClNQ)$_2$] (**6**). In this paper, we present synthesis of these complexes and the results of infrared and UV-Vis spectroscopy, and elemental and monocrystal X-ray structural analysis. Moreover, we also discuss their antiproliferative activity against seven cancer cell lines and, using one non-cancerous Cos-7 cell line, we evaluate their selectivity. We also compare their anticancer activity with the activity of the corresponding ligands and cisplatin. Finally, we present the antimicrobial activity of the complexes and ligands against one gram-positive and one gram-negative bacteria, as well as their antioxidant activity.

## 2. Results and Discussion

### 2.1. Syntheses

The described copper complexes were synthesised by a simple mixing and stirring of solutions of corresponding ligand and copper(II) salt at laboratory (**1a**) or higher temperature (**1b**–**6**). While HClQ and HBrQ ligands used in the syntheses of complexes were obtained commercially, the HClNQ, HdNQ and HClBrQ ligands were first synthesised by previously described synthetic routes [26–28]. Interestingly, in the synthesis of **1a**, where CuBr$_2$ was used, in situ bromination of HClQ ligand was observed. The mechanism of bromination of organic substances using CuBr$_2$ was described in the literature [29,30]. This motivated us to prepare HClBrQ ligand and use it for the synthesis of **1b** to compare its structure with the structure of **1a**, because, in the case of in situ bromination, only 85% of the ligand molecules were brominated, as confirmed by semiquantitative EDS analysis (Figure S1) and X-ray structural analysis. Crystals of all prepared complexes suitable for X-ray structural analysis were obtained by slow crystallisation from corresponding solutions.

The composition of the prepared complexes was suggested by elemental (**1b**–**4**, **6**) and X-ray structural analysis (**1a**, **1b**, **3**, **5**, **6**). Crystals of **5** were unstable in air, due to the releasing of DMF from the structure, and crystals of **1a** were not prepared in sufficient quantity, and, therefore, they could not be characterised by elemental analysis.

### 2.2. Infrared Spectroscopy

All complexes were first characterised by IR spectroscopy (Figure 1) to confirm the presence of the ligands or solvates in the complexes. The presence of XQ ligands in the prepared compounds was confirmed by several bands, including very weak bands of

$\nu$(C–H)$_{ar}$ vibrations observed at 3075–3098 cm$^{-1}$. Coordination of the ligands to the copper central atom was supported by the absence of $\nu$(O–H) vibration, from the hydroxyl group, which should be observed in uncoordinated 8-hydroxyquinoline and its derivatives as a broad band in the 3700–3400 cm$^{-1}$ region [31–35]. Characteristic bands of halogen functional groups in positions 5 and 7 presented, in the ranges 973–984 ($\nu$(C$_5$–Cl) vibrations in (**1a**, **1b**, **2**, **6**) and 861–873 cm$^{-1}$ ($\nu$(C$_7$–Br) vibrations in (**1a**–**3**). The presence of a nitro group in **4**–**6** was manifested by bands of $\nu$(N–O)$_{as}$ vibrations observed at 1561–1569 cm$^{-1}$ and $\nu$(N–O)$_{sym}$ vibrations at 1323 cm$^{-1}$ [35].

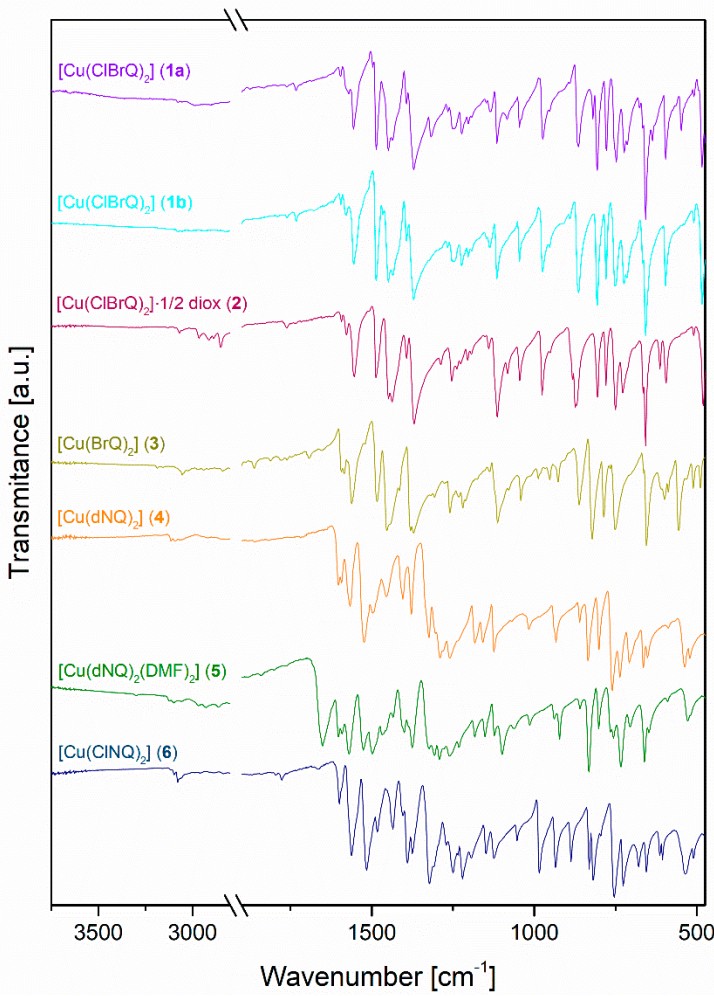

**Figure 1.** FT–IR spectra of **1**–**6**.

If we compare the IR spectra of **2** and **5** with the spectra of **1b** and **4**, respectively, we can clearly identify characteristic bands of used solvents in the IR spectra of the first two complexes. The bands of $\nu$(CH$_2$) vibrations at 2965, 2912, 2887, 2849 cm$^{-1}$ and ring breathing vibrations at 613 and 1182 cm$^{-1}$ confirmed the presence of 1,4-dioxane in **2** [36–38]. Molecules of DMF in **5** manifested themselves as weak bands of $\nu$(C–H)$_{al}$ vibrations at 2968, 2928, and 2860 cm$^{-1}$, as a band at 1098 cm$^{-1}$, which belonged to the deformation vibrations of the methyl group, and as a strong band of $\nu$(C=O) vibrations at 1651 cm$^{-1}$ [39].

### 2.3. UV-Vis Spectroscopy

A study of the stability of the prepared complexes (**1b**–**4**, **6**) was performed by comparison of the UV-Vis spectra of the complexes freshly suspended in Nujol with the spectra of the complexes in DMSO and DMSO/water (1:1) solutions, which were remeasured every 24 h over 3 days. As can be seen in Figure 2, the spectra of **1b** in DMSO and DMSO/water

measured for 3 days were identical and very similar to the spectrum of **1b** prepared in Nujol, which suggested the stability of **1b** in the solutions. Similar results were obtained for **2**, **4** and **6**, but this was not the case for **3**. Figure 2 shows its UV-Vis spectra and it was obvious that the spectra in DMSO/water did not coincide with the spectra in Nujol or DMSO. This might be explained by a very low concentration of **3** in the DMSO/water solution, due to its continuous precipitation from this solution.

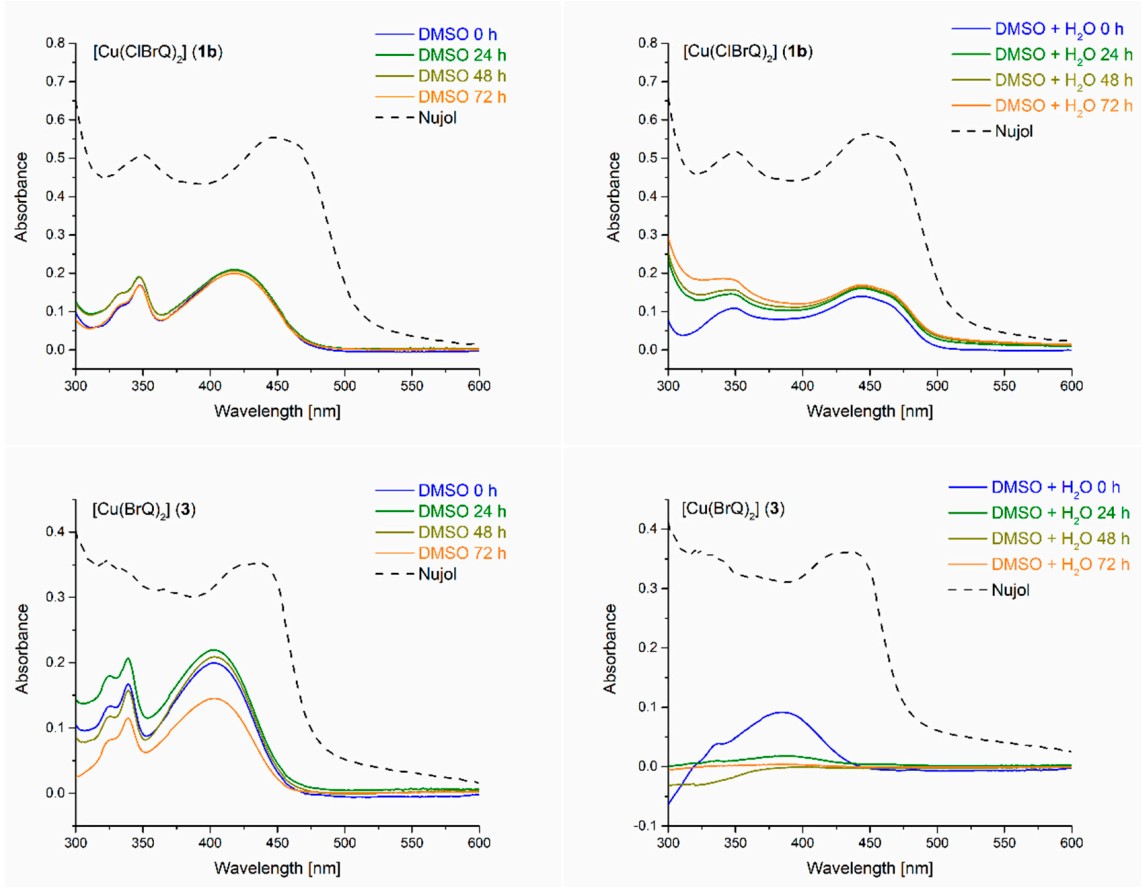

**Figure 2.** UV-VIS spectra of **1b** (**top**) and **3** (**bottom**).

### 2.4. X-ray Structure Analysis

The crystals of **1a**, **1b**, **3**, **5** and **6** were suitable for X-ray structural analysis, which confirmed their molecular character. Complexes **1a**, **1b** and **6** crystallised in the monoclinic space group $P2_1/c$, while **3** crystallised in $P2_1/n$ space group. Their unit cells contained two $[Cu(XQ)_2]$ molecules with copper atoms sitting on inversion centres. The central atoms were chelate-coordinated by two deprotonated corresponding XQ ligands (XQ = ClBrQ, BrQ and ClNQ) via oxygen and nitrogen atoms in a distorted square planar fashion (Figures 3 and S2). The shape of their polyhedral coordination was confirmed by bond lengths and angles (Table 1). Cu1–O1 bonds (1.920(2)–1.926(2) Å) were slightly shorter than Cu1–N1 bonds (1.953(2)–1.964(2) Å), due to the smaller covalent radius of the oxygen atom. Similar bond distances and angles were observed in other copper complexes with derivatives of 8-hydroxyquinoline [11,23,40,41].

Even though the crystals of **5** were not stable, being out of the maternal solution, collected X-ray data was sufficient to solve the structure. Complex **5** crystallised in the triclinic space group *P*-1. Cu1 atom, sitting on an inversion centre, was hexacoordinated by pairs of oxygen and nitrogen atoms in *trans*-positions from two molecules of dNQ, and axial positions were occupied by two oxygen atoms from two molecules of DMF (Figure 4). As can be seen in Table 2, the shape of the coordination polyhedron could be described as elongated tetragonal bipyramid, due to the Jahn–Teller effect. Nevertheless, the Cu1–O1

and Cu1–N1 bond lengths were close to those observed in **1a**, **1b**, **3** and **6**. All carbon atoms of DMF were disordered over two positions with site occupation factors being 0.62 and 0.38.

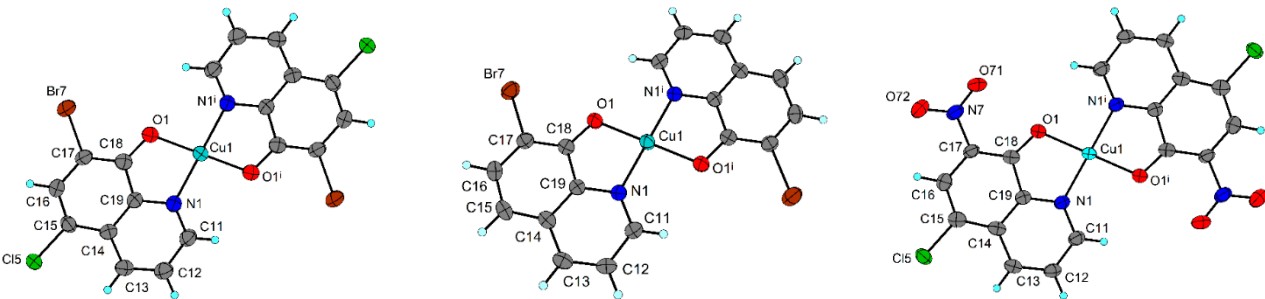

**Figure 3.** Molecular structure of **1b**, **3** and **6** (left to right). Displacement ellipsoids are drawn at the 80% or 50% (**3**) probability levels. Symmetry code: i = −*x*, −*y* + 1, −*z* + 1.

**Table 1.** Selected bond lengths [Å] and angles [°] for **1a**, **1b**, **3** and **6**.

|  | **1a** | **1b** | **3** | **6** |
|---|---|---|---|---|
| Cu1–O1 | 1.922(3) | 1.920(2) | 1.922(2) | 1.926(2) |
| Cu1–N1 | 1.964(3) | 1.965(2) | 1.958(2) | 1.953(2) |
| O1–Cu1–N1$^i$ | 94.90(11) | 94.76(8) | 94.57(9) | 95.81(9) |
| O1–Cu1–N1 | 85.10(11) | 85.24(8) | 85.43(9) | 84.19(9) |

Symmetry code: i = −*x*, −*y* + 1, −*z* + 1.

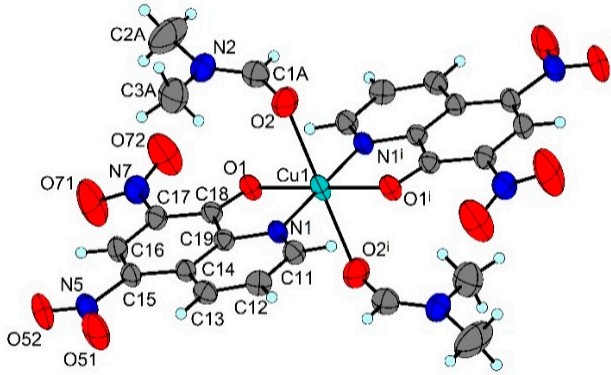

**Figure 4.** Molecular structure of **5**. Displacement ellipsoids are drawn at the 50% probability level. Only one position of carbon atoms of DMF is shown because of clarity. Symmetry code: i = −*x* + 1, −*y* + 1, −*z*.

**Table 2.** Selected bond lengths [Å] and angles [°] for **5**.

| **Bonds** |  | **Angles** |  |
|---|---|---|---|
| Cu1–O1 | 1.9508(14) | O1–Cu1–N1 $^i$ | 96.82(7) |
| Cu1–N1 | 1.9706(17) | O1–Cu1–N1 | 83.18(7) |
| Cu1–O2 | 2.4920(19) | O1–Cu1–O2 | 94.14(6) |
|  |  | N1–Cu1–O2 | 91.91(7) |
|  |  | O1$^i$–Cu1–O2 | 85.86(6) |
|  |  | N1$^i$–Cu1–O2 | 88.09(7) |

Symmetry code: i = −*x* + 1, −*y* + 1, −*z*.

From the above-described structures, only structures with nitro groups (**5** and **6**) were stabilised by hydrogen bonds. Two hydrogen bonds presented in **5** (Table S1) created a layer parallel with the (01-1) plane (Figure 5). Only one hydrogen bond in the structure of **6** (Table S1) created a layer parallel with the (100) plane (Figure 5). No other significant intermolecular interactions were present in the structures of the complexes.

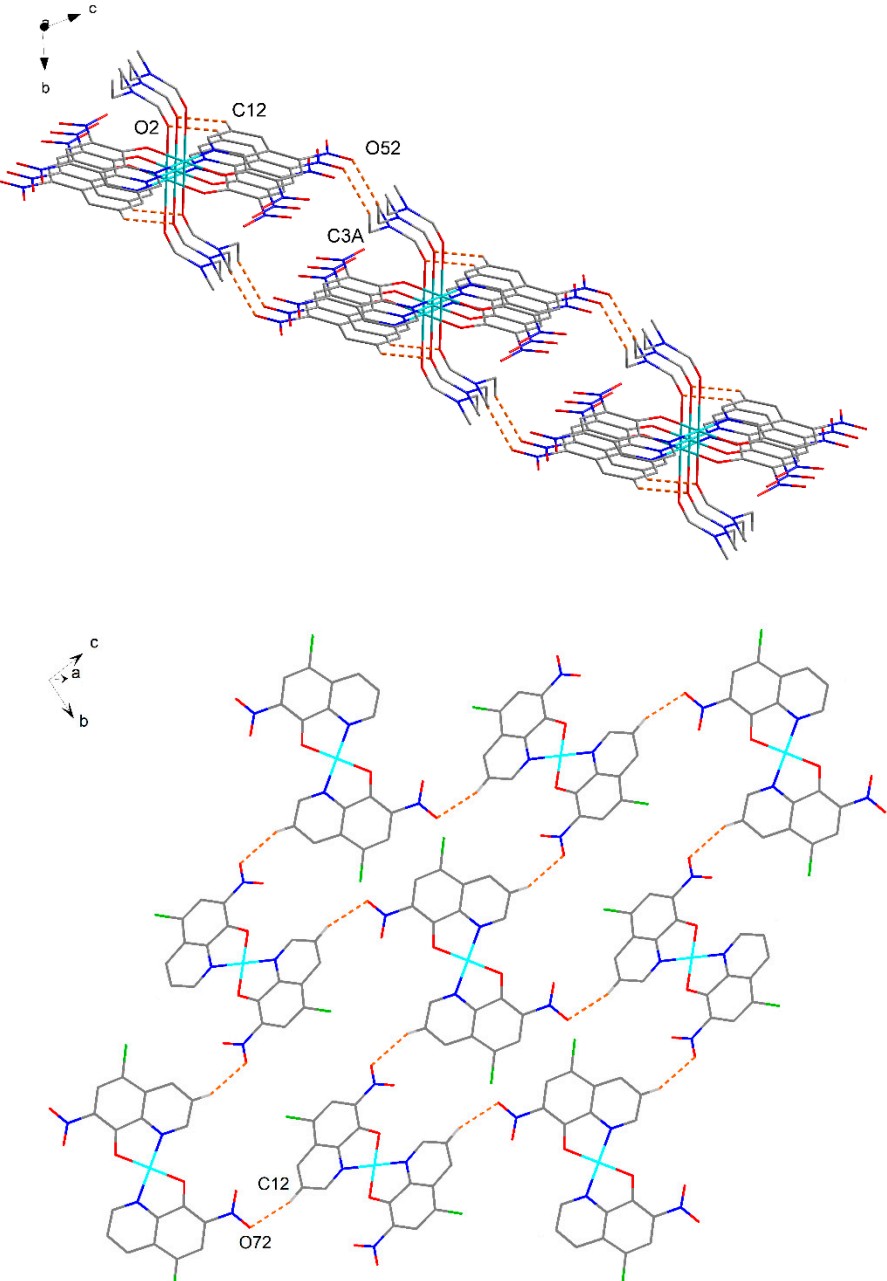

**Figure 5.** Layers formed by hydrogen bonds (dashed lines) in **5** (**top**) and **6** (**bottom**). Hydrogen atoms not involved in hydrogen bonds are omitted because of clarity.

### 2.5. Antiproliferative Activity

In the present work, four copper(II) complexes (**1b**, **3**, **4** and **6**) and their ligands were screened for potential antiproliferative activity. Furthermore, cisplatin, as a common chemotherapy agent, was used as a standard. There is evidence that copper complexes should be studied for their pro-apoptotic potential, also due to the fact that cancer cells take up larger amounts of copper than normal cells, as reviewed in [42]. Moreover, it was published that copper complexes with quinoline induced apoptosis and had cytotoxic effects on cancer cell lines [43–45]. In our study, the most potent novel copper(II) complex was **3** with $IC_{50}$ values in the range 5.3–6.0 µM on all tested cancer cell lines (Table 3). However, we did not observe selectivity towards non-cancerous Cos-7 cells ($IC_{50}$ = 5.8 µM). This complex was more efficient than cisplatin and its ligand HBrQ. Other tested complexes (**1b**, **4** and **6**) showed $IC_{50}$ values above 200 µM on all tested cell lines, except for Caco-2 cells,

where complex **1b** showed IC$_{50}$ 106.8 μM and complex **4** around 46.4 μM with a selectivity towards Cos-7 cells, but lower effectivity than cisplatin. Based on the literature [46–51] the antiproliferative activity of copper(II) chloride against the seven cancer cell lines under study was not tested, because CuCl$_2$ displayed inconsequential in vitro toxicity.

**Table 3.** IC$_{50}$ values of tested copper(II) complexes and their ligands.

|  | Cell Lines (IC50 in μM) | | | | | | | |
|---|---|---|---|---|---|---|---|---|
|  | **MCF-7** | **MDA-MB-231** | **HCT116** | **Caco-2** | **HeLa** | **A549** | **Jurkat** | **Cos-7** |
| **1a** | NT | NT | NT | NT | NT | NT | NT | NT |
| **1b** | >200 | >200 | >200 | 106.8 | >200 | >200 | >200 | >200 |
| **2** | NT | NT | NT | NT | NT | NT | NT | NT |
| **3** | 5.8 | 6.0 | 5.3 | 5.4 | 5.4 | 5.6 | 5.7 | 5.8 |
| **4** | 204.8 | >200 | >200 | 46.4 | >200 | >200 | >200 | >200 |
| **5** | NT | NT | NT | NT | NT | NT | NT | NT |
| **6** | >200 | >200 | >200 | >200 | >200 | >200 | >200 | >200 |
| HClBrQ | 36.4 | 6.2 | 23.1 | 19.7 | 40.2 | 24.4 | 5.7 | 65.3 |
| HBrQ | 9.1 | 6.1 | 5.6 | 6.5 | 20.1 | 6.4 | 5.9 | 6.5 |
| HdNQ | 78.1 | 79.5 | 81.5 | >200 | >200 | >200 | 72.8 | >200 |
| HClNQ | 6.8 | 13.0 | 8.1 | 75.8 | 47.1 | 28.0 | 34.2 | 112.2 |
| cisplatin | 29.7 | 7.1 | 7.4 | 23.2 | 35.4 | 13.5 | 6.3 | 18.3 |

NT: not tested.

*2.6. Antibacterial Activity*

The antibacterial activity of the prepared complexes and their ligands were tested against gram-positive (*S. aureus*) and gram-negative (*E. coli*) bacteria (CuCl$_2$ was not tested, due to its inactivity against bacteria [52]). The RIZD, as well as MIC, were performed. Only ligand HClNQ in RIZD exhibited test inhibition against gram-positive bacteria *S. aureus* as well as gram-negative bacteria *E. coli*. The RIZD for *S. aureus* was 153% and for *E. coli* 123% (Table 4), whereas the inhibition of gentamicin sulfate as positive control was 100%. Other complexes were not suitable for antibacterial activity in concentration of 33.6 μM. This concentration was still possible, due to the solubility of the tested compounds.

**Table 4.** Antibacterial activity of tested complexes. RIZD (%) means percentage of relative inhibition zone diameter. All complexes were tested against *E. coli* and *S. aureus* in 3 replicates (n = 3, ±SD).

| RIZD (%) | **1b** | **3** | **4** | **6** | **HClBrQ** | **HBrQ** | **HdNQ** | **HClNQ** |
|---|---|---|---|---|---|---|---|---|
| *E. coli* | NA | NA | NA | NA | NA | NA | NA | 123.51 |
| *S. aureus* | NA | NA | NA | NA | NA | NA | NA | 153.31 |

NA: no activity.

MIC was tested with ligand HClNQ in all dilutions (1:1; 1:2; 1:4; 1:8). For *E. coli*, as well as *S. aureus*, the absorbance in dilutions 1:1 and 1:2 was lower and comparable with negative control (average 0.042 ± 0.005). Dilutions 1:4 and 1:8 showed higher absorbance, due to turbidity caused by the growing bacteria (Table 5).

**Table 5.** MIC (minimal inhibition concentration) of tested complex HClNQ with dilution ratio. Values mean absorbance. Absorbance was based on the cloudiness of the sample. Higher value means that bacteria were growing. Positive control represents bacterial growth without any treatment.

|  | **1:1** | **1:2** | **1:4** | **1:8** |
|---|---|---|---|---|
| *E. coli* | | | | |
| HClNQ | 0.047 ± 0.001 | 0.051 ± 0.001 | 0.105 ± 0.007 | 0.166 ± 0.043 |
| Positive control | 0.363 ± 0.007 | 0.363 ± 0.007 | 0.363 ± 0.007 | 0.363 ± 0.007 |
| *S. aureus* | | | | |
| HClNQ | 0.046 ± 0.001 | 0.063 ± 0.003 | 0.107 ± 0.001 | 0.224 ± 0.012 |
| Positive control | 0.341 ± 0.035 | 0.341 ± 0.035 | 0.341 ± 0.035 | 0.341 ± 0.035 |

### 2.7. Radical Scavenging Activity

Radical scavenging activity was tested for complexes **1b**, **3**, **4** and **6**, along with free parental ligands against ABTS and DPPH radicals. Ligands HClBrQ and HdNQ showed low radical scavenging activity, while HClNQ was completely inactive within the measured range. Among the prepared compounds only **3** was active against both radicals (Table 6); however, its parental ligand HBrQ showed even stronger antioxidant activity [53]. The lower activity of studied complexes in comparison with free ligand molecules suggested that the radical scavenging mechanism involved reaction of radicals with 8-hydroxyquinoline derivatives, rather than Cu(II) central atoms. Antioxidant activity was more pronounced against ABTS radical than DPPH radical for all active compounds. This trend was previously observed for analogous metal complexes with the derivatives of 8-quinolinol [54,55].

**Table 6.** ABTS and DPPH radical scavenging activity ($IC_{50}$ in μM; and SC% for 200 μM antioxidant concentration) for complex **3**, free ligands and L-ascorbic acid.

|  | ABTS | | DPPH | |
|---|---|---|---|---|
|  | $IC_{50}$ (μM) | SC (%) | $IC_{50}$ (μM) | SC (%) |
| **3** | 33.33 ± 0.31 | 100 | - | 6.09 ± 2.08 |
| HClBrQ | 109.29 ± 1.50 | 82.80 ± 1.87 | - | 24.08 ± 0.78 |
| HBrQ [a] | 8.23 ± 0.52 | 100 | 46.03 ± 1.09 | 100 |
| HdNQ | - | 0 | - | 13.489 ± 0.87 |
| L-ascorbic acid | 21.03 ± 0.30 | 100 | 35.24 ± 1.25 | 100 |

Complexes **1b**, **4** and **6** and ligand HClNQ were inactive in the measured range 0–200 μM. [a] Data from [53].

## 3. Materials and Methods

### 3.1. Materials and Chemicals

The coordination compounds to be investigated were prepared using copper(II) chloride dihydrate, p.a. (Lachema, Neratovice, Czech Republic), copper(II) bromide, 99% (Sigma Aldrich, Bratislava, Slovakia), 5-chloro-8-hydroxyquinoline (HClQ), 95% (Sigma Aldrich), 7-bromo-8-hydroxyquinoline (HBrQ), 97% (Sigma Aldrich), N,N-dimethylformamide, 99% (Merck KGaA, Darmstadt, Germany), 1,2-dimethoxyethane, 99% (Alfa Aesar, Karlsruhe, Germany), ethanol, 96% (BGV, Hniezdne, Slovakia), methanol, p.a. (Centralchem, Bratislava, Slovakia), 1,4-dioxane, 99% (Centralchem), dimethyl sulfoxide, ≥99.9% (Sigma Aldrich). All commercially available chemicals were used without further purification. HdNQ, HClNQ and HClBrQ ligands were synthesised.

### 3.2. Syntheses

#### 3.2.1. Synthesis of [Cu(ClBrQ)$_2$] (**1a**)

The HClQ (35.9 mg, 0.2 mmol) was dissolved in DMF (10 mL). While continuously stirring, 10 mL of DMF solution of CuBr$_2$ (44.7 mg, 0.2 mmol) was added. After 30 min of stirring, the beaker was laid down at room temperature. After three months, yellow needles of **1a** had formed, and were filtered off, and dried in the air.

[Cu(ClBrQ)$_2$] (**1a**)—Calc. for $C_{18}H_{8.30}Br_{1.70}Cl_2N_2O_2Cu$ (554.85 g·mol$^{-1}$): C, 38.96; H, 1.51; N, 5.05%. Found: not measured. IR (ATR, cm$^{-1}$): $\nu$(C–H)$_{ar}$ 3075 (vw), $\nu$(C=C)$_{ar}$ 1595 (w), 1578 (w), 1555 (m), 1485 (m), $\nu$(C=N) 1448 (m), $\nu$(C–C) 1370 (s), 1223 (m), 1135 (m), $\nu$(C–O) 1114 (m), $\beta$(C–H) 1045 (m), $\nu$(C$_5$–Cl) 973 (m), $\nu$(C$_7$–Br) 864 (m), $\gamma$(C–H) 806 (m), Ring breathing 779 (m), 748 (m), $\beta$(CCC) 723 (m), 654 (s), $\beta$(CNC) 596 (m), $\beta$(C$_5$–Cl) 508 (w), $\gamma$(CCC) 485 (m).

#### 3.2.2. Synthesis of [Cu(ClBrQ)$_2$] (**1b**)

HClBrQ (25.9 mg, 0.1 mmol) was dissolved in ethanol (10 mL) and warmed to 60 °C. While continuously stirring, 10 mL of DMF solution of CuCl$_2$ (8.5 mg of CuCl$_2$·2H$_2$O, 0.05 mmol) (warmed to 60 °C) was added. After 30 min of stirring, the beaker was laid

down at room temperature. After five days, yellow needles of **1b** had formed, and were filtered off, and dried in the air.

[Cu(ClBrQ)$_2$] (**1b**)—Calc. for C$_{18}$H$_8$Br$_2$Cl$_2$N$_2$O$_2$Cu (578.73 g·mol$^{-1}$): C, 37.37; H, 1.39; N, 4.84%. Found: C, 37.72; H, 1.58; N, 4.65%. IR (ATR, cm$^{-1}$): $\nu$(C–H)$_{ar}$ 3074 (vw), $\nu$(C=C)$_{ar}$ 1594 (w), 1578 (w), 1555 (m), 1485 (m), $\nu$(C=N) 1448 (m), $\nu$(C–C) 1370 (s), 1223 (m), 1136 (m), $\nu$(C–O) 1114 (m), $\beta$(C–H) 1045 (m), $\nu$(C$_5$–Cl) 974 (m), $\nu$(C$_7$–Br) 863 (m), $\gamma$(C–H) 806 (m), Ring breathing 779 (m), 750 (m), $\beta$(CCC) 723 (m), 654 (s), $\beta$(CNC) 596 (m), $\beta$(C$_5$–Cl) 509 (w), $\gamma$(CCC) 484 (m).

### 3.2.3. Synthesis of [Cu(ClBrQ)$_2$]·1/2 Diox (**2**)

HClBrQ (25.9 mg, 0.1 mmol) was dissolved in 1,4-dioxane (10 mL) and warmed to 60 °C. While continuously stirring, 10 mL ethanol solution of CuCl$_2$ (8.5 mg of CuCl$_2$·2H$_2$O, 0.05 mmol) (warmed to 60 °C), was added. After 30 min of stirring, the beaker was laid down and the precipitate of **2** formed during stirring was filtered off. Mother liquor was laid to crystallise at room temperature. After five days, yellow needles of **2** had formed, and were filtered off, and dried in the air. The identity of the powder and crystals was verified by IR spectroscopy.

[Cu(ClBrQ)$_2$]·1/2 diox (**2**)—Calc. for C$_{20}$H$_{12}$Br$_2$Cl$_2$N$_2$O$_3$Cu (622,58 g·mol$^{-1}$): C, 38.58; H, 1.94; N, 4.50%. Found: C, 38.34; H, 1.92; N, 4.42%. IR (ATR, cm$^{-1}$): $\nu$(C–H)$_{ar}$ 3079 (vw), $\nu$(CH$_2$) 2965 (w), 2912 (w), 2887 (w), 2849 (m), $\nu$(C=C)$_{ar}$ 1592 (w), 1577 (w), 1553 (m), 1486 (m), $\nu$(C=N) 1448 (m), $\nu$(C–C) 1369 (s), 1253 (m), 1139 (m), $\nu$(C–O) 1113 (m), $\beta$(C–H) 1045 (m), $\nu$(C$_5$–Cl) 975 (m), $\nu$(C$_7$–Br) 873 (m), $\gamma$(C–H) 805 (m), Ring breathing 779 (m), 613 (w), 750 (m), 728 (m), $\beta$(CCC) 657 (s), $\beta$(CNC) 594 (m), $\beta$(C$_5$–Cl) 509 (w), $\gamma$(CCC) 482 (m).

### 3.2.4. Synthesis of [Cu(BrQ)$_2$] (**3**)

HBrQ (22.4 mg, 0.1 mmol) was dissolved in 10 mL of ethanol and warmed to 60 °C. While continuously stirring, 10 mL of 1,4-dioxane solution of CuCl$_2$ (CuCl$_2$·2H$_2$O of 8.5 mg, 0.05 mmol) (warmed to 60 °C) was added. After 30 min of stirring, the beaker was laid down at room temperature. After five days, yellow needles of **3** had formed, and were filtered off, and dried in the air.

[Cu(BrQ)$_2$] (**3**)—Calc. for C$_{18}$H$_{10}$Br$_2$Cl$_2$N$_2$O$_2$Cu (509.64 g·mol$^{-1}$): C, 42.42; H, 1.98; N, 5.50%. Found: C, 42.57; H, 2.10; N, 5.37%. IR (ATR, cm$^{-1}$): $\nu$(C–H)$_{ar}$ 3054 (vw), $\nu$(C=C)$_{ar}$ 1584 (m), 1561 (s), 1483 (s), $\nu$(C=N) 1453(s), $\nu$(C–C) 1371 (s), 1259 (m), 1231 (m), 1219 (m), $\nu$(C–O) 1112 (m), $\beta$(C–H) 1041 (m), $\nu$(C$_7$–Br) 861 (m), $\gamma$(C–H) 821 (m), Ring breathing 786 (m), 751 (s), $\beta$(CCC) 654 (s), $\beta$(CNC) 599 (w), 587(w), $\beta$(C–O) 555 (m), $\gamma$(CCC) 482 (m).

### 3.2.5. Synthesis of [Cu(dNQ)$_2$] (**4**)

HdNQ (23.5 mg, 0.1 mmol) was dissolved in methanol (10 mL) and warmed to 60 °C. While continuously stirring, solution of CuCl$_2$ (8.5 mg of CuCl$_2$·2H$_2$O, 0.05 mmol) in dimethoxyethane (10 mL, warmed to 60 °C) was added. After 30 min of stirring, the precipitate of **4** was filtered off and dried in the air.

[Cu(dNQ)$_2$] (**4**)—Calc. for C$_{18}$H$_8$N$_6$O$_{10}$Cu (531,84 g·mol$^{-1}$): C, 40.65; H, 1.52; N, 15.80%. Found: C, 41.04; H, 1.88; N, 15.48%. IR (ATR, cm$^{-1}$): $\nu$(C–H)$_{ar}$ 3056 (vw), $\nu$(C=C)$_{ar}$ 1594 (m), 1518 (m), 1492 (m), $\nu$(N–O)$_{as}$ 1567 (m), $\nu$(C=N) 1453 (m), $\nu$(C–C) 1405 (w), 1379 (m), 1255 (m), $\nu$(N–O)$_{sym}$ 1321 (m), $\beta$(C–H) 1169 (m), 1155 (m), 1016 (w), $\nu$(C–O) 1113 (m), $\delta$(NO$_2$) 928 (m), 699 (m), $\gamma$(C–H) 831 (m), 797 (m), Ring breathing 749 (s), 733 (w), $\beta$(CCC) 655 (m), $\beta$(CNC) 592 (w), $\beta$(C–O) 530 (m), $\gamma$(CCC) 457 (m).

### 3.2.6. Synthesis of [Cu(dNQ)$_2$(DMF)$_2$] (**5**)

HdNQ (23.5 mg, 0.1 mmol) was dissolved in DMF (10 mL) and warmed to 60 °C. While continuously stirring, 10 mL of DMF solution of CuCl$_2$ (8.5 mg of CuCl$_2$·2H$_2$O, 0.05 mmol) (warmed to 60 °C) was added. After 30 min of stirring, the beaker was laid

down at room temperature. After two days, green prisms of **5** had formed, and were filtered off, and dried in the air.

[Cu(dNQ)$_2$DMF$_2$] (**5**)—Calc. for C$_{18}$H$_8$N$_6$O$_{10}$Cu (531.84 g·mol$^{-1}$): C, 40.65; H, 1.52; N, 15.80%. Found: not measured, IR (ATR, cm$^{-1}$): $\nu$(C–H)$_{ar}$ 3058 (vw), $\nu$(C–H)$_{al}$ 2968 (vw), 2928 (vw), 2860 (vw), $\nu$(C=O) 1651 (s), $\nu$(C=C)$_{ar}$ 1601 (m), 1591 (m), 1525 (m), 1497 (m), $\nu$(N–O)$_{as}$ 1569 (m), $\nu$(C=N) 1454 (m), $\nu$(C–C)$_{ar}$ 1399 (m), 1374 (m), 1260 (m), $\nu$(N–O)$_{sym}$ 1323 (m), $\beta$(C–H) 1182 (m), 1151 (m), $\nu$(C–O) 1122 (m), $\delta$(CH$_3$) 1098 (m), $\delta$(NO$_2$) 922 (m), 705 (m), $\gamma$(C–H) 831 (m), 801 (m), Ring breathing 756 (m), 733 (s), $\beta$(CCC) 660 (m), $\beta$(CNC) 591 (w), $\beta$(C–O) 527 (m), $\gamma$(CCC) 470 (w), 450 (w).

### 3.2.7. Synthesis of [Cu(ClNQ)$_2$] (**6**)

HClNQ (22.5 mg, 0.1 mmol) was dissolved in 10 mL of DMF and warmed to 60 °C. While continuously stirring, 10 mL of DMF solution of CuCl$_2$ (8.5 mg of CuCl$_2$·2H$_2$O, 0.05 mmol) (warmed to 60 °C), was added. After 30 min of stirring, the beaker was laid down and the precipitate of **6**, that formed during stirring, was filtered off. The mother liquor was laid to crystallise at room temperature. After five days, green needles of **6** had formed, and were filtered off, and dried in the air. The identity of the powder and crystals was verified by IR spectroscopy.

[Cu(ClNQ)$_2$] (**6**)—Calc. for C$_{18}$H$_8$Cl$_2$N$_4$O$_6$Cu (510.73 g·mol$^{-1}$): C, 42.50; H, 1.78; N, 10.97%. Found: C, 42.33; H, 1.58; N, 10.97%. IR (ATR, cm$^{-1}$): $\nu$(C–H)$_{ar}$ 3078 (vw), $\nu$(C=C)$_{ar}$ 1599 (m), 1482 (s) $\nu$(N–O)$_{as}$ 1561 (m), $\nu$(C=N) 1434 (s), $\nu$(C–C) 1384 (m), 1374 (m), 1249 (m), 1220 (s), $\nu$(N–O)$_{sym}$ 1323 (s), $\nu$(C–O) 1123 (m), $\beta$(C–H) 1148 (m), 1052 (m), $\nu$(C$_5$–Cl) 984 (m), $\delta$(NO$_2$) 934 (m), 679 (m), $\gamma$(C–H) 831 (m), 818 (s), Ring breathing 754 (s), $\beta$(CCC) 726 (m), 679 (m), 655 (s), $\beta$(CNC) 613 (w), $\beta$(C–O) 535 (m), $\beta$(C$_5$–Cl) 510 (w), $\gamma$(CCC) 476 (m).

### 3.3. Physical Measurements

Infrared spectra of prepared complexes were recorded on a Nicolet 6700 FT-IR spectrometer from Thermo Scientific with a diamond crystal Smart OrbitTM, in the range 4000–400 cm$^{-1}$. Elemental analyses of C, H, and N were with a CHNS Elemental Analyzer varioMICRO from Elementar Analysensysteme GmbH. Absorption spectra were measured with a SPECORD 250 spectrophotometer (Analytik Jena, Jena, Germany), from 300 to 600 nm, in Nujol, and DMSO and DMSO/water (1:1) solutions at 24, 48 and 72 h intervals. IR and UV-Vis spectra were described in Origin 2022b [56]. The morphological characteristics of the samples were studied using a JEOL JSM 6510 scanning electron microscope (W cathode, 20 nm resolution at 1 kV). Semi-quantitative chemical analysis of the samples was determined using the attached Oxford Instruments EDS analyser INCA X act. Measurements were performed on native samples without any conductive coating.

### 3.4. X-ray Structure Analysis

The crystal structures of **3** and **5** were determined using an Oxford Diffraction Xcalibur2 diffractometer equipped with a Sapphire2 CCD detector, while the structures of **1a**, **1b** and **6** were determined using a Rigaku XtaLAB Synergy S diffractometer with Hybrid Pixel Array detector (HyPix-6000HE). CrysAlis Pro software was used for data collection and cell refinement, data reduction and absorption correction [57]. Structures of prepared complexes were solved by SHELXT [58] and refined by SHELXL [59], implemented in the WinGX program [60]. For all non-H atoms, anisotropic displacement parameters were refined. Hydrogen atoms of XQ and DMF molecules were placed in calculated positions and refined riding on carbon atoms. Presence of hydrogen bonds was analysed by using SHELXL, while PLATON [61], running under WinGX, was used to analyse π–π interaction. Diamond was used for molecular graphics [62]. The summary of crystal data and structure refinements for **1a**, **1b**, **3**, **5** and **6** is presented in Table 7.

**Table 7.** Crystal data and structural refinement of **1a**, **1b**, **3**, **5** and **6**.

| Compound | 1a | 1b | 3 | 5 | 6 |
|---|---|---|---|---|---|
| Empirical formula | $C_{18}H_{8.30}Br_{1.70}Cl_2CuN_2O_2$ | $C_{18}H_8Br_2Cl_2CuN_2O_2$ | $C_{18}H_{10}Br_2CuN_2O_2$ | $C_{24}H_{22}CuN_8O_{12}$ | $C_{18}H_8Cl_2CuN_4O_6$ |
| Formula weight [g·mol$^{-1}$] | 554.85 | 578.52 | 509.64 | 678.03 | 510.72 |
| Temperature [K] | 100(2) | 100(2) | 173(2) | 173(2) | 100(2) |
| Wavelength [Å] | 1.54184 | 1.54184 | 0.71073 | 0.71073 | 1.54184 |
| Crystal system | monoclinic | monoclinic | monoclinic | triclinic | monoclinic |
| Space group | $P2_1/c$ | $P2_1/c$ | $P2_1/n$ | $P-1$ | $P2_1/c$ |
| Unit cell dimensions [Å, °] | $a = 4.89650(10)$ | $a = 4.88650(10)$ | $a = 4.9383(2)$ | $a = 6.5539(6)$ | $a = 3.7670(2)$ |
|  | $b = 10.32300(10)$ | $b = 10.3474(3)$ | $b = 10.1335(4)$ | $b = 10.1953(7)$ | $b = 12.4567(6)$ |
|  | $c = 17.5282(2)$ | $c = 17.5221(4)$ | $c = 16.2276(6)$ | $c = 11.1513(7)$ | $c = 18.1614(5)$ |
|  | $\beta = 90.182(1)$ | $\beta = 90.201(2)$ | $\beta = 90.581(3)$ | $\alpha = 103.833(5)$ | $\beta = 93.452(3)$ |
|  |  |  |  | $\beta = 100.445(6)$ |  |
|  |  |  |  | $\gamma = 103.446(7)$ |  |
| Volume [Å$^3$] | 885.99(2) | 885.96(4) | 812.02(5) | 681.18(9) | 850.67(7) |
| Z; density (calculated) [g·cm$^{-3}$] | 2; 2.080 | 2; 2.169 | 2; 2.084 | 1; 1.653 | 2; 1.994 |
| Absorption coefficient [mm$^{-1}$] | 9.162 | 9.962 | 6.280 | 0.883 | 5.193 |
| $F(000)$ | 538 | 558 | 494 | 347 | 510 |
| Crystal shape, colour | needle, yellow | needle, yellow | needle, yellow | prism, green | needle, green |
| Crystal size [mm] | 0.161 × 0.078 × 0.031 | 0.050 × 0.040 × 0.020 | 0.560 × 0.228 × 0.064 | 0.654 × 0.216 × 0.152 | 0.048 × 0.015 × 0.009 |
| $\theta$ range for data collection [°] | 4.972–77.225 | 4.964–76.881 | 3.216–28.707 | 3.295–28.514 | 4.306–77.467 |
| Index ranges | $-5 \leq h \leq 6$, $-13 \leq k \leq 13$, $-21 \leq l \leq 22$ | $-5 \leq h \leq 6$, $-12 \leq k \leq 11$, $-21 \leq l \leq 21$ | $-6 \leq h \leq 6$, $-13 \leq k \leq 11$, $-20 \leq l \leq 21$ | $-8 \leq h \leq 8$, $-13 \leq k \leq 13$, $-11 \leq l \leq 14$ | $-2 \leq h \leq 4$, $-11 \leq k \leq 15$, $-22 \leq l \leq 22$ |
| Reflections collected/independent | 30,356/1833 [$R$(int) = 0.0494] | 8341/1769 [$R$(int) = 0.0323] | 5221/1863 [$R$(int) = 0.0257] | 7356/3093 [$R$(int) = 0.0261] | 6520/1709 [$R$(int) = 0.0335] |
| Data/restrains/parameters | 1833/0/124 | 1769/0/124 | 1863/0/115 | 3093/0/237 | 1709/0/142 |
| Goodness-of-fit on $F^2$ | 1.161 | 1.062 | 1.070 | 1.089 | 1.161 |
| Final $R$ indices [$I > 2\sigma(I)$] | R1 = 0.0347, wR2 = 0.0768 | R1 = 0.0271, wR2 = 0.0776 | R1 = 0.0312, wR2 = 0.0646 | R1 = 0.0428, wR2 = 0.0905 | R1 = 0.0367, wR2 = 0.1057 |
| $R$ indices (all data) | R1 = 0.0371, wR2 = 0.0777 | R1 = 0.0296, wR2 = 0.0788 | R1 = 0.0441, wR2 = 0.0718 | R1 = 0.0549, wR2 = 0.0975 | R1 = 0.0452, wR2 = 0.1093 |
| Largest diff. peak and hole [e·Å$^{-3}$] | 0.435 and −0.476 | 0.458 and −0.573 | 0.424 and −0.384 | 0.402 and −0.452 | 0.461 and −0.643 |

### 3.5. Cell Cultures

The human cancer cell lines were purchased from ATCC (American Type Culture Collection; Manassas, VA, USA). HCT116 (human colorectal carcinoma), HeLa (human cervical adenocarcinoma) and Jurkat (human leukemic T cell lymphoma) were cultured in RPMI 1640 medium (Biosera, Kansas City, MO, USA) while A549 (human alveolar adenocarcinoma), MCF-7 (human Caucasian breast adenocarcinoma), Caco-2 (human colorectal adenocarcinoma) and MDA-MB-231 (human breast cancer cell line) were maintained in a growth medium consisting of high glucose Dulbecco's Modified Eagle Medium (DMEM) + sodium pyruvate (Biosera). The human kidney fibroblasts (Cos-7) were cultured in DMEM medium (Biosera). All media were supplemented with a 10% fetal bovine serum (FBS; Invitrogen, Carlsbad, CA, USA), Antibiotic/Antimycotic Solution (Sigma, St. Louis, MO, USA) and maintained in an atmosphere containing 5% $CO_2$ in humidified air at 37 °C. Prior to each experiment, cell viability was greater than 95%.

### Screening of Antiproliferative/Cytotoxic Activity

The antiproliferative/cytotoxic effect of copper complexes (concentrations of 10, 50 and 100 μM) was determined by resazurin assay in HCT116, Caco-2, A549, MCF-7, MDA-MB-231, HeLa, Jurkat and Cos-7 cells. Tested cells ($1 \times 10^4$/well) were seeded in 96-well plates. After 24 h, final copper complex concentrations, prepared from DMSO stock solution, were added, and incubation proceeded for the next 72 h. Ten microliters of resazurin solution (Merck, Darmstadt, Germany), at a final concentration of 40 μM, was added to each well at the end-point (72 h). After a minimum of 1 h incubation, the fluorescence of the metabolic product resorufin was measured by the automated Cytation$^{TM}$ 3 cell imaging multi-mode reader (Biotek, Winooski, VT, USA) at 560 nm excitation/590 nm emission filter. The results were expressed as a fold of the control, where control fluorescence was taken as 100%. All experiments were performed in triplicate. The $IC_{50}$ values were calculated from these data.

### 3.6. Antibacterial Activity

#### 3.6.1. Microorganisms Used

The tested bacteria (S. aureus CCM 4223 and E. coli CCM 3988) were obtained from the Czech collection of microorganisms (CCM, Brno, Czech Republic).

#### 3.6.2. Agar Well-Diffusion Method

The antibacterial properties of the four complexes, **1b**, **3**, **4** and **6**, and their ligands, HClBrQ, HBrQ, HdNQ and HClNQ, were evaluated by the agar well diffusion method using a slightly modified process compared to [63]. Firstly, each compound was dissolved in a small amount of 100% DMSO and then dissolved to 33.6 μM solution. As a positive control, gentamicin sulfate (Biosera, Nuaille, France), with concentration 50 μg/mL, was used.

Bacteria were cultured overnight, aerobically, at 37 °C in LB medium (Sigma-Aldrich, Saint-Louis, MO, USA), with agitation. The inoculum from these overnight cultures was prepared by adjusting the density of the culture to equal that of the 0.5 McFarland standard ($1$–$2 \times 10^8$ CFU/mL), by adding a sterile saline solution. These bacterial suspensions were diluted 1:300 in liquid plate count agar (HIMEDIA, Mumbai, India) resulting in a final concentration of bacteria approximately $5 \times 10^5$ CFU/mL, and 20 mL of this inoculated agar was poured into a Petri dish (diameter 90 mm). Once the agar was solidified, five mm diameter wells were punched in the agar and filled with 50 μL of samples. Gentamicin sulfate, with a concentration of 50 μg/mL, was used as a positive control. The plates were incubated for 18–20 h at 37 °C. Afterwards, the plates were photographed, and the inhibition zones were measured by ImageJ 1.53e software (U. S. National Institutes of Health, Bethesda, MD, USA). The values used for the calculation were mean values calculated from 3 replicate tests.

The antibacterial activity was calculated by applying the formula reported in [63]:

$$\%RIZD = [(IZD\ sample - IZD\ negative\ control)/IZD\ gentamicin] \times 100$$

where RIZD is the relative inhibition zone diameter (%) and IZD is the inhibition zone diameter (mm).

As a negative control, the inhibition zones of 5% DMSO equal to 0 were taken. The inhibition zone diameter (IZD) was obtained by measuring the diameter of the transparent zone.

### 3.6.3. Determination of the Minimum Inhibitory Concentration (MIC) by the Microdilution Method

The minimum inhibitory concentration was determined by the microdilution method with a slight modification of the procedure described in [64].

Stock solutions of the test samples prepared in 5% DMSO (Sigma-Aldrich, USA) were two-fold diluted (1:1 to 1:8) in wells of a 96-well plate (Greiner Bio-One, Germany): the wells of the microtiter plate were filled with 50 μL of Mueller–Hinton Broth (MHB, HIMEDIA, Mumbai, India), and 50 μL of stock solution of the test substance was added to the first well (1:1 dilution). After mixing, 50 μL of this solution was transferred to the next well (1:2 dilution), etc. Each sample was tested in triplicate.

Bacteria were cultured overnight, aerobically, at 37 °C in MHB, with agitation. The inoculum from the overnight cultures was prepared by adjusting the density of bacterial suspension to 0.5 McFarland standard ($1–2 \times 10^8$ CFU/mL) by adding sterile saline and then diluting 1:150 in MHB. Subsequently, 50 μL of this inoculum was added to each well with 50 μL of diluted test samples (final concentration of bacteria in the well ca. $5 \times 10^5$ CFU/mL). Wells filled only with MHB medium and bacterial suspension were used as a positive control, and wells filled with sterile MHB alone were used as a negative control. The plates were covered with a lid and incubated for 18–20 h at 37 °C. The evaluation was made by measuring the absorbance at 600 nm on a microplate reader (Synergy HT, Biotek, Santa Clara, CA, USA).

### 3.7. Radical Scavenging Experiments

Radical scavenging activity of complexes and free ligands was estimated by DPPH (2,2-diphenyl-1-picrylhydrazyl) and ABTS (diammonium 2,2′-azino-bis(3-ethylbenzothiazoline-6-sulfonate) radicals, according to slightly modified methods described in literature [65,66]. Compounds were dissolved in DMSO and the resulting solutions were diluted with MeOH in 1:3 ratio. The prepared solutions were mixed with methanolic solutions of the respective radicals and incubated in the dark at room temperature (7 min ABTS and 30 min DPPH). The absorbance was recorded at 734 nm (ABTS) and 517 nm (DPPH), respectively. L-ascorbic acid was used as a standard, and experiments were performed in triplicate. The $IC_{50}$ parameters were calculated from a linear plot of the inhibition percentage against the concentration of compounds.

### 4. Conclusions

In this work, we present six new copper(II) complexes: $[Cu(ClBrQ)_2]$ (**1a**, **1b**), $[Cu(ClBrQ)_2] \cdot 1/2$ diox (**2**) (diox = 1,4-dioxane), $[Cu(BrQ)_2]$ (**3**), $[Cu(dNQ)_2]$ (**4**), $[Cu(dNQ)_2(DMF)_2]$ (**5**) and $[Cu(ClNQ)_2]$ (**6**), where HClBrQ is 5-chloro-7-bromo-8-hydroxyquinoline, HBrQ is 7-bromo-8-hydroxyquinoline, HClNQ is 5-chloro-7-nitro-8-hydroxyquinoline and HdNQ is 5,7-dinitro-8-hydroxyquinoline. Prepared complexes were characterised by IR spectroscopy (**1–6**), and elemental (**1b–4** and **6**) and X-ray structural (**1a**, **1b**, **3**, **5** and **6**) analysis. The sability of **1b–4** and **6** in DMSO and DMSO/water solutions was verified by UV-Vis spectroscopy.

Complexes **1a**, **1b**, **3**, **5** and **6** were molecular compounds. Crystal structures of **1a**, **1b**, **3** and **6** formed by square planar $[Cu(XQ)_2]$ complexes, in which copper atoms were coordinated by two pairs of oxygen and nitrogen atoms from two deprotonated XQ ligands. On the other hand, a tetragonal bipyramidal coordination of the copper atom in **5** was observed. The equatorial plane was occupied by the same atoms as in the mentioned $[Cu(XQ)_2]$ complexes, and oxygen atoms from DMF molecules occupied axial positions. Interestingly, despite similar structures, intramolecular interactions were observed only in

**5** and **6**, where hydrogen bonds existed. They created layers parallel with (01-1) (**5**) and (100) (**6**) plane.

After cytotoxic evaluation, only complex **3** showed promising potential as an anti-cancer agent, but, due to the low selectivity towards non-cancerous cells, modification of the complex was required.

Among the prepared compounds only **3** was active against both tested radicals (ABTS and DPPH). Antioxidant activity could be involved in the cytotoxic effects of **3** against the tested cell line; however, the free ligand showed even better antiradical properties. Apparently, the coordination of HBrQ ligand to Cu(II) decreased its antioxidant activity.

Only the HClNQ ligand showed antibacterial potential with concentration 33.6 μM.

**Supplementary Materials:** The following supporting information can be downloaded at: https://www.mdpi.com/article/10.3390/inorganics10120223/s1, Figure S1: SEM micrograph (a) of **1a** along with EDS elemental analysis (b); Figure S2: Molecular structure of **1a**; Table S1: Hydrogen bonds [Å and °] for **5** and **6**.

**Author Contributions:** Conceptualization, I.P. and M.K. (Martina Kepeňová); investigation, M.K. (Martina Kepeňová), M.K. (Martin Kello), R.S., M.G., R.F., Ľ.T., M.L., J.Š. and I.P.; resources, M.K. (Martin Kello), M.G., M.L. and I.P.; writing—original draft preparation, M.K. (Martina Kepeňová), M.K. (Martin Kello), R.S., M.G. and I.P.; Writing—Review & Editing, M.K. (Martina Kepeňová), R.S. and I.P.; visualization, M.K. (Martina Kepeňová) and J.Š.; supervision, M.K. (Martin Kello), M.G. and I.P.; project administration, M.K. (Martin Kello), M.G. and I.P.; funding acquisition, M.K. (Martin Kello), M.L. and I.P. All authors have read and agreed to the published version of the manuscript.

**Funding:** This work was supported by VEGA 1/0148/19, VEGA 1/0653/19, APVV-18-0016 and VVGS-PF-2022-2134. Moreover, this publication is the result of the project implementation, "Open scientific community for modern interdisciplinary research in medicine (OPENMED)", ITMS2014+: 313011V455, supported by the Operational Programme Integrated Infrastructure, funded by the ERDF. We also thank the Research Infrastructure NanoEnviCz project, supported by the Ministry of Education, Youth and Sports of the Czech Republic, Project No. LM2018124 for instrumentation.

**Data Availability Statement:** CCDC 2216977-2216981 contain the supplementary crystallographic data for **1a**, **1b**, **3**, **5**, and **6**. These data can be obtained free of charge via http://www.ccdc.cam.ac.uk/conts/retrieving.html, (accessed on 5 November 2022) or from the Cambridge Crystallographic Data Centre, 12 Union Road, Cambridge CB2 1EZ, UK; Fax: (+44) 1223-336-033; or e-mail: deposit@ccdc.cam.ac.uk.

**Acknowledgments:** Special thanks go to Petra Ecorchard. (Centre of Instrumental Techniques, Institute of Inorganic Chemistry of the CAS) for the preparing of SEM samples.

**Conflicts of Interest:** The authors declare no conflict of interest. The funders had no role in the design of the study; in the collection, analyses, or interpretation of data; in the writing of the manuscript; or in the decision to publish the results.

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
