# Peer review of "Low-Dimensional Compounds Containing Bioactive Ligands. Part XIX: Crystal Structures and Biological Properties of Copper Complexes with Halogen and Nitro Derivatives of 8-Hydroxyquinoline"

_inorganics, doi:10.3390/inorganics10120223_

Round 1

Reviewer 1 Report

This paper describes the syntheses, characterizations, crystal structures of Cu(II) complexes with 8-hydroxyquinoline derivatives and antitriproliferative, antibacterial, and radical scavenging activities were investigated. The findings of this study are interesting. However, the identification of complex 1a is insufficient (see additional comments). Thus, this paper is worth publishing in Inorganics with minor revisions. Some additional comments are listed below.

1) X-ray crystal structure of 1a should be solved according to the chemical composition obtained by elemental analysis. The SEM result gives only local composition of the surface of the sample.

2) page 9, line 356: What is the precipitate? I expect this is a hydrated complex 3.

Reviewer 2 Report

The manuscript 'Low-dimensional compounds containing bioactive ligands. Part XIX: Crystal structures and biological properties of copper complexes with halogen and nitro derivatives of 8-hydroxyquinoline' is a continuation of a long series of papers devoted to the synthesis, study of properties and study of biological properties of d metal complexes with different ligands. These papers contribute to coordination chemistry of d metal ions and create a basis for further use of the complexes in medicine. The present manuscript is decent, and I have only a couple of minor comments:

1. The similarity in IC50 values observed for complex 3 and corresponding ligand HBrQ clearly shows that it is free ligand that has a biological effect rather than complex. Furthermore, it is confirmed by the UV-Vis spectral observation (Fig. 2) showing the dissociation of complex 3 in the solution. I would add this information. This conclusion is further confirmed by known fact that metal complexes tend to dissociate in the presence of biological macromolecules [10.1016/S0162-0134(00)00129-X, 10.1016/j.saa.2018.08.009].

2. It seems to me that the most plausible mechanism of radical scavenging activity is not "radical scavenging", technically. Only ascorbic acid takes ROS and undergoes oxidation forming dehydroascorbic acis [10.1038/s41598-021-86477-8, 10.3390/inorganics10070102]. Other ligands rather bind into complexes the traces of metal ions that participate in the Fenton reaction (Fe2+, Cu2+). Complexed metals are unable to catalyze ROS formation [10.1023/A:1013348005312].

I would consider adding these ideas to the paper, and that's it.

Reviewer 3 Report

The paper of Ivan Potočňák and co-authors is a fundamental work on synthesis new Copper(II) compounds containing biological active 8-hydroxyquinoline halogen derivatives ligands. The authors obtained six new complexes, where crystal structures for four of them had been determined (and one polymorph).

According to “Functionalization of Molecular Architectures: Advances and Applications on Low-Dimensional Compounds” by Kazuhiro Shikinaka, ISBN: 9781315150697 and some published papers  

https://doi.org/10.1021/acsnano.1c03836

https://doi.org/10.1002/asia.201301586

https://doi.org/10.1039/C2DT31243C

Low-dimensional compounds are a concept for molecules that corresponds with rod, ladder (one-dimensional), and sheet (two-dimensional)-shaped (macro) molecules. The most well-known low-dimensional molecules are carbon nanotube and graphene.

As we saw from the article, the authors are dealing with simple monoionic (mononuclear) copper(II) complexes. Perhaps the authors confuse these concepts. I have read previous articles titled the same way - the same mistake is made there. Low-dimensional compounds are valuable precisely in crystalline (or solid) phase. Why discuss the structure in a crystal of the obtained compounds if the authors use them from article to article in the study of biological properties in solution? In solution, there will most likely be partial dissociation with the formation of ion pairs without any long-range order. How to name the article is up to the authors, but I think that the authors make a mistake.

Line 117 (and further): “The identity of the powder and crystals was verified by IR spectroscopy”. To prove the correspondence between the crystalline phase and the powder phase, it is necessary to make X-Ray powder diffraction. The coincidence of the IR spectra only indicates the presence of the same types of bonds, but says nothing (or little) about the structure. For example, compounds 1a and 1b have different structures in the crystal, but their IR spectra are almost identical (as shown in the experimental part). How can we be sure that the product 1b is phase pure and does not contain impurity of 1a? Impurities in biologically active substances and drugs must be controlled.

The biological activity studies lack the activity of copper(II) chloride against the same cell lines and bacteria. No need to measure, add literature data from databases, e.g. reaxys. Since powder diffraction was not carried out for the obtained samples, we cannot speak about the absence of copper chloride impurity in the powders of substances.

Line 321: “crystals of 1a were not prepared in sufficient amount” Have there been any attempts to make this reaction in an inert atmosphere?

In general, the manuscript is well written and designed, there is enough data for publication in Inorganics (5 structures, six complexes). The current work seems interesting to me and I recommend it to publish in Inorganics as a part of Special Issue "Recent Progress in Coordination Chemistry".
